# Loss of a Functional Mitochondrial Pyruvate Carrier in *Komagataella phaffii* Does Not Improve Lactic Acid Production from Glycerol in Aerobic Cultivation

**DOI:** 10.3390/microorganisms11020483

**Published:** 2023-02-15

**Authors:** Ana Caroline de Oliveira Junqueira, Nadielle Tamires Moreira Melo, Nádia Skorupa Parachin, Hugo Costa Paes

**Affiliations:** 1Department of Molecular Biology, University of Brasília, Brasília 70790-900, Brazil; 2Genomic Sciences and Biotechnology, Catholic University of Brasília, Brasília 70790-160, Brazil; 3Gingko Bioworks, 27 Drydock Avenue, 8th floor, Boston, MA 02210-2377, USA; 4Clinical Medicine Division, University of Brasília Medical School, University of Brasília, Brasília 70910-900, Brazil

**Keywords:** mitochondrial pyruvate carrier 1, bacterial hemoglobin, L-leucine, qPCR, PLA

## Abstract

Cytosolic pyruvate is an essential metabolite in lactic acid production during microbial fermentation. However, under aerobiosis, pyruvate is transported to the mitochondrial matrix by the mitochondrial pyruvate carrier (MPC) and oxidized in cell respiration. Previous reports using *Saccharomyces cerevisiae* or *Aspergillus oryzae* have shown that the production of pyruvate-derived chemicals is improved by deleting the *MPC1* gene. A previous lactate-producing *K. phaffii* strain engineered by our group was used as a host for the deletion of the *MPC1* gene. In addition, the expression of a bacterial hemoglobin gene under the alcohol dehydrogenase 2 promoter from *Scheffersomyces stipitis*, known to work as a hypoxia sensor, was used to evaluate whether aeration would supply enough oxygen to meet the metabolic needs during lactic acid production. However, unlike *S. cerevisiae* and *A. oryzae*, the deletion of Mpc1 had no significant impact on lactic acid production but negatively affected cell growth in *K. phaffii* strains. Furthermore, the relative quantification of the *VHb* gene revealed that the expression of hemoglobin was detected even in aerobic cultivation, which indicates that the demand for oxygen in the bioreactor could result in functional hypoxia. Overall, the results add to our previously published ones and show that blocking cell respiration using hypoxia is more suitable than deleting Mpc for producing lactic acid in *K. phaffii*.

## 1. Introduction

The development of metabolically engineered microorganisms represents an opportunity to replace chemicals in the petrochemical industry with biobased or renewable resources. To transform microorganisms into competitive cell factories, genetic adjustments are required and could improve the use of alternative carbon sources and enable the production of relevant metabolites. 

The yeast *Komagataella phaffii* is often used for the heterologous expression of peptides and proteins [1,2] with several biotechnological applications, such as the production of humanized antibodies [3]. In addition to its well-established characteristics as its high secretion capacity and post-translational modifications, the ability to use methanol [4] and glycerol [5,6] as the sole carbon source turns it into a promising host for the construction of cell factories based on the valorization of crude glycerol. 

Crude glycerol is the main byproduct of the biodiesel industry, formed by a mixture of glycerol and impurities [7]. The incentives and subsidies to increase the use of biofuels in many countries underlie the projection that biodiesel production will increase to 55 billion L/year in 2031 [8]. However, during the transesterification reaction, oil is mixed with alcohol, usually methanol, yielding biodiesel and approximately 10% crude glycerol [7]. The presence of methanol in crude glycerol narrows its potential uses due to its toxicity, but for *K. phaffii,* it could be used as an additional carbon source. In the glycerol assimilation pathway, glycerol enters the payoff phase of glycolysis after two reactions (Figure 1). For this reason, the production of platform chemicals derived from pyruvate, such as lactic acid, is promising in glycerol valorization using *K. phaffii*. 

Lactic acid exhibits light acidity (pKA = 3.85), high hygroscopicity, and thermal stability. Altogether, these characteristics enable its application in the food and beverages industry (e.g., as an acidulant), pharmaceutical industry (e.g., as an antimicrobial agent), and healthcare industry (e.g., in chemical peeling) [9]. Indeed, the commercial value of lactic acid was estimated at around 1.1 billion US dollars in 2020 and is estimated to reach 2.1 billion by 2025 [10]. However, with the growing demand for biocompatible materials in recent years, lactic acid has been used to produce polylactide (PLA) [10], a thermostable and biocompostable bioplastic [11]. PLA can be used to make food packaging, containers, cups, plates, and filaments for 3D printing [9,10].

Several strategies were used to increase the productivity of lactic acid in genetically engineered yeasts and fungi. These include the overexpression of monocarboxylate transporters such as Jen1 and Ady2 [12,13], disruption of native lactate dehydrogenases (e.g., *CYB2*) to prevent lactate consumption [14], modifications to correct the redox balance such as the disruption of the *NDE1/NDE2* external NADH dehydrogenase genes [15], and changes to improve or enable the consumption of alternative carbon sources [16,17]. However, lactic acid production depends on the availability of pyruvate in the cytosol. Thus, it is essential to target competing consumption routes to avoid the diversion of pyruvate to them, such as by disrupting pyruvate decarboxylase (*PDC*) genes [18,19].

The cytosolic pyruvate from glycolysis crosses the mitochondrial outer membrane through porins. It is then transported through the mitochondrial inner membrane by a transporter known as the mitochondrial pyruvate carrier (Mpc) [20]. The Mpc was identified in 1971 from rat mitochondria [21], but its molecular identity was only identified in 2012 [22,23]. The heterocomplex is well conserved among mammals, *Drosophila*, and yeasts, and can be formed by Mpc1/Mpc2 or Mpc1/Mpc3 [22,23]. In the yeast *Saccharomyces cerevisiae*, the Mpc complex varies accordingly to the culture conditions, as Mpc1 with Mpc2 complex under fermentative conditions and Mpc1 with Mpc3 in respiratory conditions [24]. Mitochondrial pyruvate is oxidized in cell respiration, so knocking out the Mpc complex in aerobic microorganisms can enable lactic acid production under aerobiosis. For example, a recent study on the filamentous fungus *Aspergillus oryzae* has demonstrated the synergistic effect of the double deletion of *PDC1* and *MPC1* in lactic acid production in aerobiosis [25]. 

In previous work by our group, the expression of a bovine lactate-dehydrogenase gene enabled the production of lactic acid from glycerol in *K. phaffii* [12]. Indeed, the disruption of the pyruvate decarboxylase (*PDC1*) in *K. phaffii* to reduce the shift of pyruvate to acetate formation resulted in the strain GLp (GS115: pGAP-LDH^+^ Δ*pdc1 His^+^*) which yielded 64.6% of lactic acid from glycerol under hypoxia [19]. However, persistent arabitol production was observed in lactate-producing strains of *K. phaffii* under hypoxia, probably due to an NAD^+^/NADH imbalance resulting from the lack of Pdc1 [26]. 

In this work, we explored the pyruvate redirection to lactic acid by the double deletion of *PDC1* and *MPC1* in a lactate-producing strain of *K. phaffii* under aerobiosis. In addition, to evaluate whether aeration would supply enough oxygen for our *K. phaffii* strain during lactic acid production, the gene of the bacterial hemoglobin from *Vitreoscilla stercoraria* (VHb) [27] was cloned under the alcohol dehydrogenase 2 promoter from *Scheffersomyces stipitis* (*SsADH2*), known to work as a hypoxia sensor [28]. A previous study using a *K. phaffii* showed that the expression of the *VHb* gene cloned under *SsADH2* promoter was 24-fold higher in low oxygen condition (>5% dissolved oxygen) compared to high oxygen [28]. 

The previously engineered GLp strain was used as a host for Mpc1 disruption resulting in the strain GLpm (Figure 1).

## 2. Materials and Methods

### 2.1. Construction of the Strains

The L-lactate-producing *K. phaffii* GLp strain described by Melo et al. 2018 [19] was used as the parent strain for this work. The identification of the putative gene encoding the subunit 1 of the mitochondrial pyruvate carrier (*MPC1*) in the genome of *K. phaffii* was done in NCBI. The search revealed an ORF in chromosome 1 for the hypothetical protein under the accession number XM_002490794.1 located in the complementary strand (Figure 2). The deletion of *MPC1* used a synthetic construct with 700 bp of the 3’ UTR from the leading strand right before the end of the gene (PAS_FragB_0028, Figure 2) and 700 bp of the 5′ UTR before the initiation codon of *MPC1* (PAS_FragB_0030, Figure 2) as flanking regions to guide homologous recombination. The construct was composed of two expression cassettes: a selection marker cassette with the gene of the hygromycin phosphotransferase (Figure 2, B. *Hyg^R^*) under the promoter of peroxisomal malate dehydrogenase [29]; and a cassette harboring the hemoglobin gene from β-proteobacteria *Vitreoscilla stercoraria* codon optimized for *K. phaffii* under the alcohol dehydrogenase 2 promoter [30]. The cassette was synthesized by GenScript (Piscataway, NJ, USA).

*K. phaffii* transformation was done according to Wu and Letchwork [31]. Transformants were selected in YPD (1% yeast extract, 2% peptone, and 2% dextrose) agar plates supplemented with 0.2 mg/mL hygromycin B (Sigma-Aldrich, St. Louis, MO, USA). Integration into the correct locus and the correct gene deletion was confirmed by PCR using primers listed in Table 1. 

### 2.2. Assessment of Growth in Rich, Complex, and Defined Media

The first assessment of the impact of Mpc1 deletion on *K. phaffii* was done by comparing the growth curves and the specific growth rate (μ = h^−1^) in three conditions: on a rich complex medium, a defined medium without amino acids, and a defined medium with leucine supplementation. The rich complex medium YPD (1% yeast extract, 2% peptone, and 2% dextrose) and the synthetic defined medium UAB were used. The preparation of UAB was described previously [19]. A stock solution of 600 mg L^−1^ of leucine was used to supplement UAB (final concentration, 6 mg L^−1^). GLpm and GLp strains were inoculated at an initial optical density (OD at 600 nm) of 0.1 in 50 mL of medium in 500 mL baffled shake flasks and incubated at 30 °C and 200 rpm for approximately 62 h. Samples of 0.2 mL were taken periodically for cell density measurement in a spectrophotometer. 

### 2.3. Bioreactor Cultivations

#### 2.3.1. Pre-Inoculum

One loopful of frozen stock (−80 °C) of yeast was used to streak YPD agar plates (YPD with 1.5% agar) supplemented with 0.1 mg/mL Zeocin (ThermoFisher; the resistance marker used to introduce the bovine L-LDH gene during the construction of the GLp strain in ref. [11]). After incubation at 30 °C, a single colony was inoculated into 50 mL of the appropriate medium (YPD or UAB) with 2% glycerol or dextrose and incubated at 30 °C and 200 rpm.

#### 2.3.2. Cultivation of GLp in the Presence of an Mpc Inhibitor

The GLp strain was cultured in 150 mL of UAB medium with 6% (*v*/*v*) glycerol in a 1.3-L bioreactor (BioFlo, Eppendorf, Germany) at pH 5.0 and 30 °C, starting at an OD_600_ of 0.5. The compound 2-Cyano-3-(1-phenyl-1H-indol-3-yl)-2-propenoic acid, commercially known as UK-5099 (PZ0160-5MG, Sigma-Aldrich), known to inhibit the activity of the mitochondrial pyruvate carrier [32], was added to the culture after 15 h at a concentration of 20 μM. Samples of 1 mL were then collected periodically for further analysis. Experiments were conducted in duplicate.

#### 2.3.3. Small-Scale Batch Cultivation

Cultures were carried out in a 250 mL vessel DASbox^®^ mini bioreactor system (Eppendorf, Germany) using 180 mL of YP media with 4% (*v*/*v*) glycerol at pH 5.0 and 28 °C. Dissolved oxygen (DO) concentration was set to 50% and controlled by adjusting the stirrer speed from 400 to 1000 rpm, and inlet compressed air flow from 10.8 to 16 L h^−1^. The automated injection of 15% (*v*/*v*) ammonia solution controlled pH at 5.0. The starting OD_600_ was 0.05. Samples (1 mL) were taken periodically for further analysis. Experiments were conducted in duplicates.

#### 2.3.4. Batch and Fed-Batch in Medium-Scale Cultivations

The defined medium UAB was used for batch and fed-batch cultivations in a 3 L bioreactor BioFlo 115 (New Brunswick, Eppendorf, Edison, NJ, USA). Batch and fed-batch cultures started with an initial OD_600_ of 0.5 at 30 °C, and pH was maintained at 5.0 with a 15% ammonia solution. The dissolved oxygen concentration was set to 50% and controlled by adjusting the stirrer speed from 300 to 900 rpm and 2 vvm. 

In batch cultivation, glycerol was added once at 6% (*v*/*v*). The culture started with 2% (*v*/*v*) glycerol in fed-batch cultivation. After 72 h of the inoculation time, cultures were fed in a single pulse for a final concentration of 10% (*v*/*v*) glycerol.

### 2.4. Quantitation of Metabolites on HPLC

Samples collected from cultivations were centrifuged at 12,100× *g* for 5 min at room temperature. After centrifugation, the supernatant was diluted five-fold and filtered using a 0.22 µm filter into 1.5 mL vials for HPLC analysis.

A Rezex ROA-Organic Acid H^+^ (8%) LC Column (300 × 7.8 mm, 00H-0138-K0, Phenomenex) was used for the HPLC (Shimadzu, Kyoto, Japan) measurements. A solution of 5 mM H_2_SO_4_ was used as the mobile phase in isocratic mode at a flow rate of 0.6 mL/min at 60 °C. Lactate, acetate, and pyruvate concentrations were measured with a UV/VIS detector at 210 nm (SPD-20A, Shimadzu, Kyoto, Japan), and glycerol, arabitol, and ethanol concentrations were measured with a refractive index detector (RID-10A, Shimadzu, Kyoto, Japan). Samples were analyzed using the software LabSolutions, version 5.54 (Shimadzu, Kyoto, Japan). 

### 2.5. Biomass Determination and Kinetic Parameters of Cultivations

Cell growth was monitored periodically in all cultures by measuring optical density at 600 nm. Dry cell weight (DCW) was used for biomass determination in bioreactor cultures using the gravimetric method in duplicates. In small-scale cultivation in rich media (DASbox^®^ mini bioreactor system), 1 mL culture samples were collected and placed in dry, pre-weighed 1.5 mL tubes and centrifuged at maximum speed for 5 min at room temperature. For each sample, the supernatant was removed, the pellet was washed with distilled water to remove the excess medium, and centrifuged again. The supernatant was removed and tubes were dried at 60 °C for 72 h. Then, the tubes were cooled in a desiccator for at least one hour and weighed. In medium-scale cultivation in UAB (BioFlo reactor), a larger volume sample (3–10 mL) was filtered through dry, pre-weighed 0.45 μm polyether sulfone filters (Frisenette, Knebel, Denmark) and washed with distilled water. Filters were dried in a microwave oven at 150 W for 20 min, cooled in a desiccator, and weighed. The DCW (g L^−1^) was the difference between the recorded weights before and after the tubes or filter. 

The kinetic parameters used to compare lactic acid-producing *K. phaffii* strain in this work were calculated according to Equations (1)–(3):μ = ln (X_2_/X_1_)/(t_2_ − t_1_),(1)
Y_x/s_ = (X*_f_* − X*_i_*)/(S*_f_* − S*_i_*) or Y_Lac/s_ = (Lac*_f_* − Lac*_i_*)/(S*_f_* − S*_i_*),(2)
P_x_ = (X*_f_* − X*_i_*)/(t*_f_* − t*_i_*) or P_Lac_ = (Lac*_f_* − Lac*_i_*)/(t*_f_* − t*_i_*),(3)
for specific growth rate (μ = h^−1^) (1), global yield (Y = g L^−1^/ g L^−1^) (2), and productivity (P = g L^−1^ h^−1^) (3); X is the biomass measured as dry cell weight per unit of volume (g L^−1^), Lac is the lactic acid concentration (g L^−1^), and S is the substrate concentration (g L^−1^). The subscripts “i” and “f” indicate the initial and final points, while subscripts “2” and ”1” indicate the measure between two consecutive time points, respectively.

### 2.6. Quantitative RT-PCR

Frozen cells *pellet* of GLpm strain collected during cultivation in a bioreactor were used for total RNA extraction. Total RNA was extracted and treated with DNase I using the Aurum™ Total RNA Mini Kit (Bio-Rad, Hercules, CA, USA). A total of 250 ng of RNA was reverse transcribed using the GoScript™ Reverse Transcriptase (Promega, Madison, WI, USA). Reaction products were diluted with nuclease-free water and amplified by PCR with the Forget-Me-Not™ EvaGreen^®^ qPCR Master Mix (Low ROX) (Biotium, Fremont, CA, USA) in a 7500 Fast Real-Time PCR System (Thermo Fisher Scientific, Waltham, MA, USA). The expression of *VHb* in GLpm samples was analyzed in triplicate using the *K. phafii ACT1* gene as a constitutive control. Oligos used are listed in Table 1. The relative quantification (RQ) of the bacterial hemoglobin (*VHb*, Figure 2, Table 1) was calculated using the −ΔΔCT method in 7500 Software v2.3 (Thermo Fisher Scientific). Oligos used are listed in Table 1.

### 2.7. Statistical Analysis

The data presented represent the average of independent experiments as the mean ± standard deviation. The differences between the specific growth rate (μ) of the parental (GLp) and transformed (GLpm) strains were compared with a *t* test followed by Tukey’s post hoc test (*p* < 0.05).

## 3. Results and Discussion

### 3.1. Effect of the Mpc Inhibitor UK-5099 on Lactic acid Production by GLp

Previous studies have thoroughly investigated the role of inactivation or overexpression of the Mpc complex in mammalian cells [20,33]. However, little research has been conducted to show the impact of engineering Mpc in yeast cells to increase the pyruvate pool in the cytosol to produce value-added metabolites. Aiming at understanding if the inhibition of the Mpc complex could increase lactic acid formation in an engineered strain of *K. phaffi*, the previously generated GLp strain was cultured in 6% (*v*/*v*) glycerol supplemented with the Mpc inhibitor UK-5099, and without leucine under aerobiosis in a 1.3 L bioreactor. The growth curve, glycerol consumption, lactic acid production, and byproduct formation are shown in Figure 3.

After 15 h of the inoculation time, the cell density reached 11.13 ± 0.53, and 3.53 ± 0.82 g L^−1^ of lactic acid was produced. At this point, 20 μM of UK-5099 was added to the bioreactor using a sterile syringe. The IC_50_ for UK-5099 adequate for blocking the pyruvate exchange by the heterodimer Mpc1/Mpc3 in *Saccharomyces cerevisiae* was determined to be 9.0 ± 7.0 μM [34]. This value is higher than the concentration of IC_50_ determined for mammalian cells (0.050 μM) [32]. Then, 20 μM was used in this experiment to ensure that Mpc would be inhibited. After 24 h of the addition of UK-5099 (39 h of cultivation time), the OD reached 84.5 ± 9.19 (21.11 ± 1.57 g L^−1^ DCW), while lactic acid and arabitol reached concentrations of 58.81 ± 12.53 and 1.3 ± 0.175 g L^−1^, respectively. This was the highest lactic acid concentration detected by our group’s engineered strain of *K. phaffii*. In previous work, the GLp strain was cultured under the same conditions without the UK-5099, resulting in 10.7 g L^−1^ of lactic acid and approximately 20 g L^−1^ of biomass [26]. This result supports the hypothesis that blocking the Mpc complex associated with the disruption of the pyruvate decarboxylase (Pdc1) could redirect pyruvate to lactic acid formation, as it showed a 5.5-fold change from the preliminary assay. However, arabitol remains the main byproduct even in aerobiosis, though its amount is reduced relative to the GLp strain [19].

### 3.2. Auxotrophy to Leucine Indicates the Lack of Mpc1 Activity 

Since the incubation of GLp with UK-5099 demonstrated the feasibility of Mpc disruption as a strategy to increase lactic acid yields, the GLpm strain was generated by integrating the deletion cassette (Figure 2), and its genotype was confirmed by PCR (Appendix A). The first phenotype observed in GLpm was poor growth compared to the parental GLp strain in a defined medium. Furthermore, 2-isopropylmalate, a precursor of L-leucine biosynthesis, depends on pyruvate metabolism. Indeed, yeasts require mitochondrial pyruvate for L-leucine and other branched-chain amino acids (BCAA) biosynthesis [35]. Thus, poor growth in synthetic media without leucine is a phenotype that indicates a lack of Mpc1 activity [22]. According to the screening described by Bricker et al. [22] and Herzig et al. [23], both the parental and the engineered strain were tested in a spot assay test (Appendix A) and in three culture media to evaluate Mpc activity, and the knockout was found to be auxotrophic for leucine. The growth curves of the parental (GLp) and the transformed (GLpm) strains are shown in Figure 4, and the specific growth rates are presented in Table 2.

In complex, rich media (YPD), the specific growth rate of GLp strain was 1.32 times higher than GLpm (Table 2). This result indicates that the lack of the *MPC1* gene impacted cell growth even in a rich medium. However, this discrepancy was accentuated when strains were grown in the synthetic defined media commonly used for *K. phaffii*, UAB. In UAB, the specific growth rate was lower relative to the rich media for both strains; however, after 22 h, the cell density (OD_600_) reached 3.55 ± 0.07 for GLp and only 0.60 ± 0.14 for GLpm. As expected, there was no significant difference between the GLp growth in UAB with or without L-leucine (Figure 2, Table 2).

Conversely, the lowest growth rate in this experiment was found in GLpm in defined media without amino acids (Table 2). Moreover, the addition of L-leucine slightly improved the growth rate of GLpm, which indicates that the deletion of MPC1 compromises the activity of the Mpc complex. This result is in accordance with the phenotype observed in Mpc1-deficient *S. cerevisiae*, which grew slowly in defined media without amino acids and recovered by adding valine and/or L-leucine [23]. Previous studies have evidence that the lack of Mpc1 is sufficient to block the activity of Mpc since it is required to form a functional heteromeric complex with Mpc2 or Mpc3 [20,22,23].

### 3.3. Lactic Acid Production in GLp and GLpm in Aerobiosis with 4% Glycerol

The GLpm strain had its genotype confirmed, and the phenotype observed in the previous test confirms the interruption of Mpc. However, the deletion of *MPC1* was previously performed in *S. cerevisiae* [36,37] and in the filamentous fungi *A. oryzae* [25] for producing metabolites derived from pyruvate, but those strains were tested in rich, complex media or excess of amino acid supplementation. For this reason, the first assay to evaluate lactic acid production in *K. phaffii* strains was conducted in a batch in YP with 4% glycerol in aerobiosis. The growth curve, concentration of metabolites, and glycerol consumption in the bioreactor are shown in Figure 5. Kinetic parameters are summarized in Table 3. 

Although there was a difference in growth rate, as shown in the previous section, metabolite curves followed a similar pattern for the parental and the engineered strains. Indeed, the titer (g L^−1^), productivity (g L^−1^ h^−1^), and yield of lactic acid were similar for both strains cultured in 4% glycerol (Table 3). This result is at odds with the effect observed in GLp in the presence of UK-5099. However, the concentration of extracellular pyruvate was higher in GLpm strains, which could indicate a pyruvate spillover due to *MPC1* deletion (Figure 5D). It must be noted that in the case of the chemical inhibition of Mpc, the drug was added after biomass generation in the culture, which is not the same situation as having the carrier absent from the beginning, which may have resulted in a metabolic adaptation of the strain. 

In engineered *A. oryzae* strains, the deletion of *MPC1* and *PDC* had a synergistic effect improving lactic acid production from glucose [25]. However, in the fermentative *S. cerevisiae*, complementary targets were more effective than the deletion of *MPC* genes for producing ethanol, 2,3-butanediol (23BD), and ethyl lactate [36,37]. For 23BD production, the deletion of *MPC1* increased 14.3-fold the concentration of the product, while targeting the *ATG32* gene responsible for inducing mitophagy resulted in a 23.3-fold increase [36]. In addition, ethyl lactate production was increased by the deletion of the *POR2* gene responsible for mitochondrial permeability, whereas the knockout of the *MPC2* had no significant effect [37]. Conversely, the overexpression of *MPC* genes has been used to increase the pyruvate pool inside the mitochondria to produce isobutanol. The biosynthetic pathway of isobutanol in *S. cerevisiae* depends on the conversion of pyruvate into 2-ketoisovalerate inside the mitochondria, and the latter is then converted into isobutyraldehyde and isobutanol [38]. The overexpression of *MPC1* and *MPC3* resulted in a 22-fold increase in isobutanol production from glucose in engineered *S. cerevisiae* strain [38].

Regarding glycerol consumption, glycerol was almost depleted in 39 h for GLp (0.58 ± 0.45 g L^−1^) and in 42 h for GLpm (0.21 ± 0.03 g L^−1^). Then, the glycerol concentration grew in both cultures, reaching final titers of 6.11 ± 0.72 g L^−1^ and 6.89 ± 0.42 g L^−1^. This can be explained by the reverse reaction of the NAD^+^-dependent glycerol 3-phosphate dehydrogenase (*GPD1*), which converts dihydroxyacetone-phosphate back to glycerol-3-phosphate [6], or of Cyb2, which can convert lactate to pyruvate while delivering electrons to the respiratory chain [14]. 

GLpm was cultivated in UAB without leucine supplementation in batch and fed-batch mode with glycerol to compare with previous cultures of GLp [19,26]. Besides the more extended lag phase, the curves presented a similar pattern compared to cultivations in 4% glycerol in rich media (Figure 5). The growth curve, metabolite concentrations, and glycerol consumption in batch and fed-batch are shown in Figure 6, and the kinetic parameters are summarized in Table 3.

The results in Figure 6 highlight the significant adaptation phase of GLpm in UAB, as it took approximately 50 h for both batch and fed-batch cultures. Interestingly, cells grew as expected after this period, reaching the cell densities of 88.8 ± 1.8 and 148 ± 8.5 in batch and fed-batch, respectively. However, compared to GLp cultivations in previous studies, there was no significant positive effect of deletion of *MPC1* on lactic acid production in *K. phaffii*. 

It is unclear why *K. phaffii* failed to show a synergistic increase in lactic acid production when both Pdc1 and Mpc1 were knocked out. A clue may be that we did not see a significant increase in acetate, or any detection of ethanol, in a strain that received the bovine *LDH* gene in a Pdc1+ background [12]. It would appear that *K. phaffii* has other possible destinations to pyruvate than its consumption by pyruvate decarboxylase. Both results could be explained if the yeast had a way to circumvent the loss of Mpc1 and shunt pyruvate carbon into the mitochondrion. Conversion to oxaloacetate by pyruvate carboxylase, which can be found in the cytosol in *S. cerevisiae* [39], followed by mitochondrial uptake via oxaloacetate transporters [40], is a possibility that merits further investigation. Indeed, a simple BLASTP search using the oxaloacetate transporter from *S. cerevisiae* (Oac1; Gene ID 853739) against the *K. phaffii* genome yields a protein that has already been annotated as being its putative orthologue in that organism, with 82% similarity (Gene ID 8197207). As for pyruvate carboxylase, a cytoplasmic isoform has been annotated in GenBank under Gene ID 8198982. 

### 3.4. Relative Expression Level of VHb

The bacterial hemoglobin from *V. stercoraria* was expressed under a promoter induced by hypoxia to gauge the demand for intracellular oxygen during batch cultivation and to test whether its presence improved lactic acid production in aerobiosis. The expression of *VHb* has already been shown to improve recombinant expression in *K. phaffii* [30,41]; however, it failed to show any improvements in lactic acid production compared with the GLp parent. The RNA of the GLpm strain was extracted from cell pellets collected during batch cultivations in 4% glycerol in YP. The cell density at three-time points and the relative quantitation of *VHb* in GLpm are shown in Figure 7.

The low solubility of molecular oxygen could challenge aerobic bioprocesses involving Crabtree-negative yeasts such as *K. phaffii*. Previous studies have demonstrated that the expression of the *VHb* gene in *K. phaffii* improved the productivity of the heterologous expression of proteins [30,41,42]. Thus, using a hypoxia-activated promoter in *K. phaffii* has already been demonstrated to result in VHb production only when the cells require intracellular oxygen [28]. The expression of *VHb* at 48 h (Figure 7) suggests that even under aerobiosis, there is a functional hypoxia resulting from lactic acid production. Given that the loss of Mpc1 would reduce the metabolic performance of the yeast under aerobiosis, as shown by our results of a slightly increased lag phase without any improvements in yield by the knockout strain, this requirement for oxygen suggests it might be challenging to use aerobiosis to improve yields in a bioreactor setting with *K. phaffii*. Thus, hypoxic conditions, beset as they are by limitations caused by a redox imbalance that prevents attaining higher yields, are still superior to the alternative for this bioprocess. Further improvements are bound to require more extensive genetic engineering and novel strategies to circumvent the existing hurdles. 

## Figures and Tables

**Figure 1 microorganisms-11-00483-f001:**
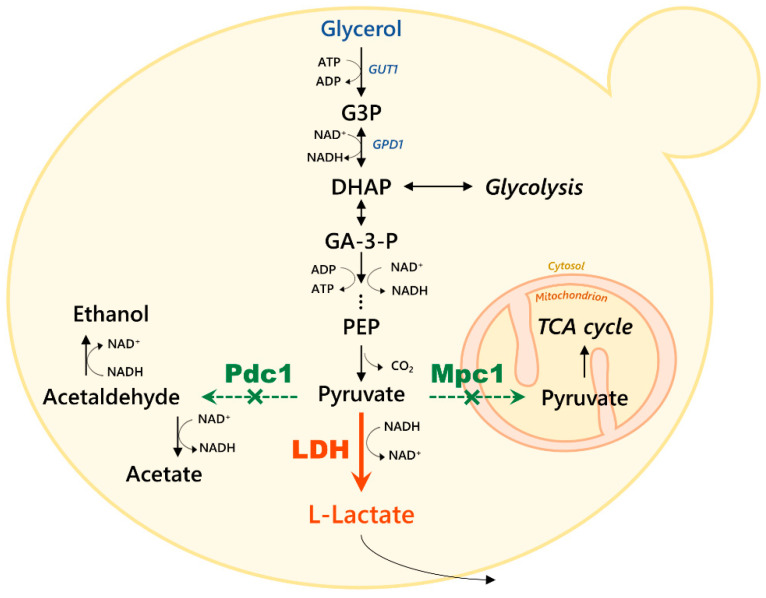
Genetic modifications of the *K. phaffii* GLpm strain. Endogenous genes of the glycerol assimilation pathway are highlighted in blue. The bovine lactate dehydrogenase is depicted in orange [12]. The targets of knockout, *PDC1* [19] and *MPC1* are shown in green. GUT1, glycerol kinase; GPD1, glycerol 3-phosphate dehydrogenase; LDH, lactate dehydrogenase; Mpc1, mitochondrial pyruvate carrier; Pdc1, pyruvate decarboxylase; G3P, glycerol-3-phosphate; DHAP, dihydroxyacetone phosphate; GA-3-P, glyceraldehyde 3-phophate; PEP, phosphoenolpyruvate.

**Figure 2 microorganisms-11-00483-f002:**
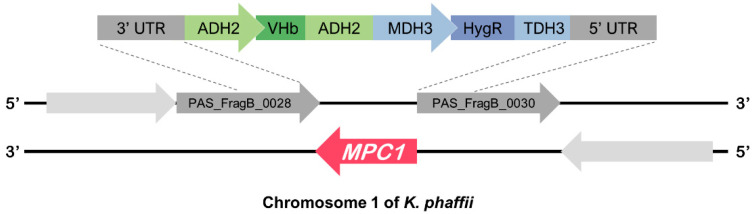
The *MPC1* gene of *K. phaffii* is located in the complementary strand of chromosome 1. The disruption cassette with 700 bp region of homology flanks the *MPC1* gene in the *K. phaffii* genome. The brown areas in both schemes are the homology flanks that drive the recombinational knock-in of the cassette into the locus.

**Figure 3 microorganisms-11-00483-f003:**
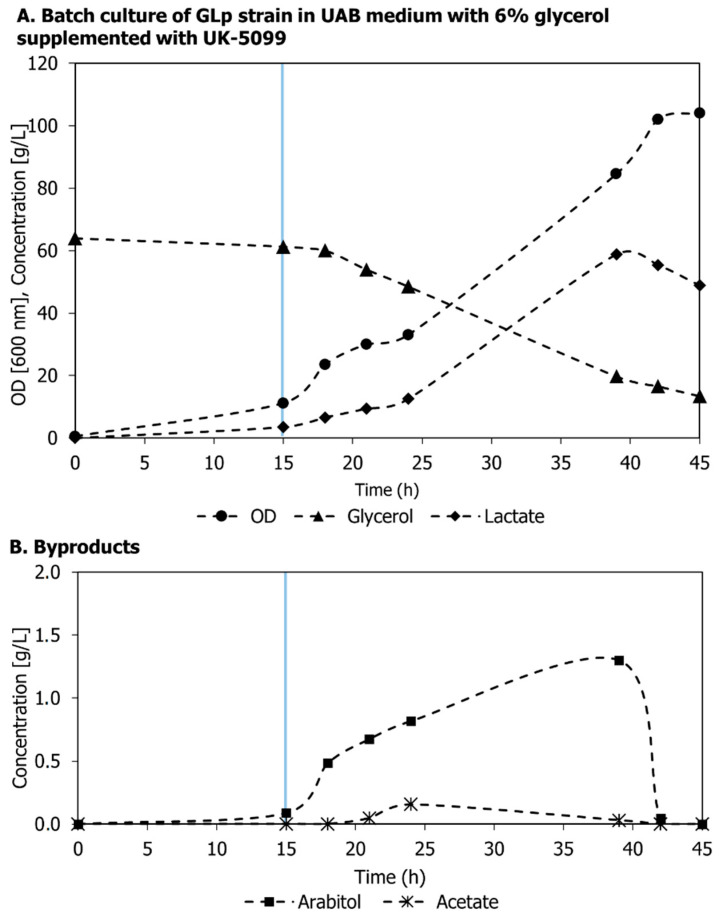
Cultivation profile of the GLp strain in 6% (*v*/*v*) glycerol supplemented with the Mpc inhibitor UK-5099 under aerobiosis in a 1.3 L bioreactor: (**A**) growth curve (circle), glycerol consumption (triangle), and lactic acid production (diamond); (**B**) arabitol (square) and acetate (asterisk) represented less than 1.5 g L^−1^. Experiments were carried out in biological duplicates. The blue line indicates the time of supplementation with UK-5099.

**Figure 4 microorganisms-11-00483-f004:**
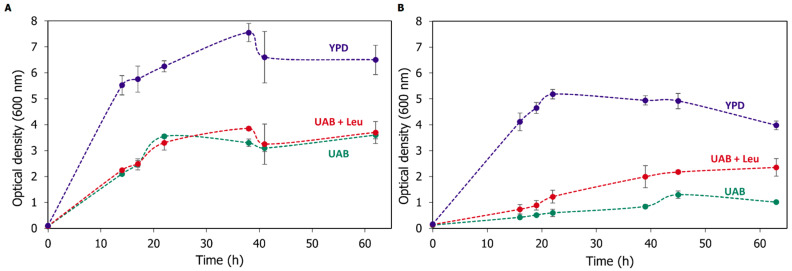
Growth curves of *K. phaffii* strains cultivated in 500 mL baffled flasks in 50 mL of YPD (blue), UAB (green), and UAB with L-leucine (red), all sourced with 2% dextrose: (**A**) GLp; (**B**) GLpm. Experiments were carried out in biological triplicates.

**Figure 5 microorganisms-11-00483-f005:**
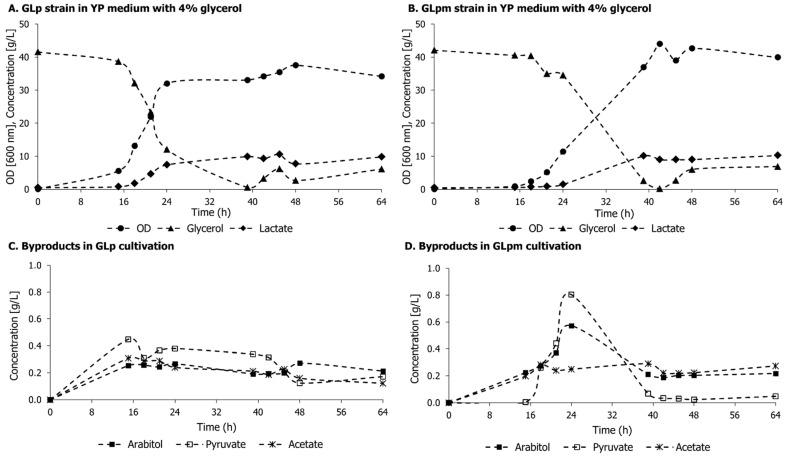
Growth curves, glycerol consumption, lactic acid production, and byproduct formation of GLp (**A**,**C**) and GLpm (**B**,**D**) strains in YP medium with 4% glycerol in the DASbox^®^ mini bioreactor system. Experiments were carried out in biological duplicates.

**Figure 6 microorganisms-11-00483-f006:**
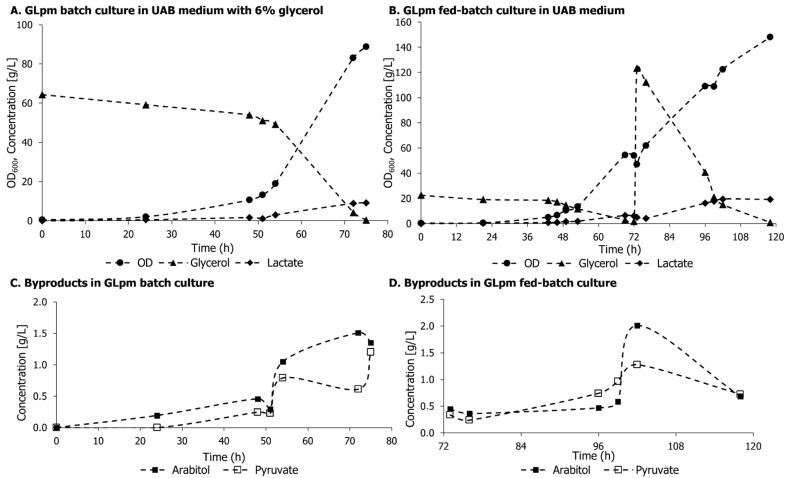
Growth curves, glycerol consumption, lactic acid production, and byproduct formation in GLpm batch (**A**,**C**) and fed-batch (**B**,**D**) cultivations in a 3 L bioreactor in UAB medium.

**Figure 7 microorganisms-11-00483-f007:**
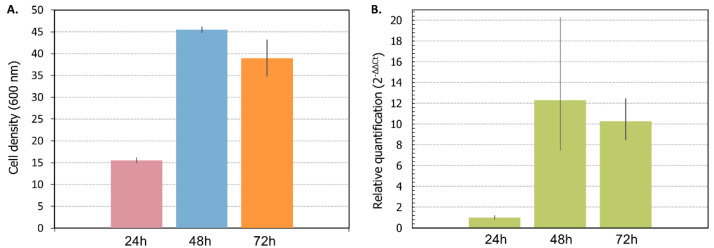
Relative quantification of the bacterial hemoglobin expressed in GLpm in batch cultivation under aerobiosis: (**A**) cell density at 24, 48, and 72 h of the cultivation time; (**B**) relative quantification of the bacterial hemoglobin (*VHb*) from RNAs extracted from cell pellets at 24, 48, and 72 h of the cultivation time. *ACT1* gene was used as endogenous control, and timepoint 24 has the reference. Bars are SDs from two experiments.

**Table 1 microorganisms-11-00483-t001:** Strain, plasmid, and primers used in this work.

Material	Description		Reference
*Strain*			
GLp	GS115: pGAP-LDH *Bos taurus* Δpdc1		[19]
GLpm	GLp: VHb *V. stercoraria* Δmpc1		This work
*Plasmid*			
	mpc-*VHb*-*Hyg^R^*		This work
*Primers*	*Sequence (5′* *→* *3′)*		
MPC1F	CTCAGATCGATAGAGTGCAAG	MPC1F with MPC1R: 543 bp amplicon in wild type and no amplicon in transformant. RAD9F with MPC1R: amplification occurs only if insertion occurs in the *locus.*	This work
MPC1R	GGAGAAGCTCCATTCGAC		This work
RAD9F	CTCTATGCCTTGAACTATGTCG		This work
qVHbF	CATCTTGCCAGCCGTTAAGAAG	Relative quantification of the heterologous hemoglobin expression.	This work
qVHbR	AACAACTCCTGACCGACGATAG		This work
qACT1F	TGTTGGTTGTCCTCGGTTGA	Constitutive control for quantitative PCR.	This work
qACT1R	TGAGCTTGGATTCGGCAGAT		This work

**Table 2 microorganisms-11-00483-t002:** The specific growth rate (μ: h^−1^) of GLp and GLpm in shake-flasks.

Strain	YP	UAB	UAB + Leucine
GLp (parental)	0.245 ± 0.008 ^a^	0.207 ± 0.005 ^b^	0.209 ± 0.008 ^b^
GLpm (Δ*mpc1*)	0.185 ± 0.000 ^c^	0.074 ± 0.000 ^d^	0.097 ± 0.001 ^e^

Means not followed by the same letter are significantly different from one another (*p*, 0.05), using the Tukey test (mean ± standard deviation). Superscript letters indicate statistical difference categories.

**Table 3 microorganisms-11-00483-t003:** Lactic acid production by the GLpm-engineered strain in the bioreactor.

Strain	Specific Growth Rate (h^−1^)	Titer of Lactic Acid (g L^−1^)	Productivity (g L^−1^ h^−1^)	Y_P/S_ (g L^−1^/g L^−1^)	Y_X/S_ (g L^−1^/g L^−1^)	Medium
GLp (parental)	0.308 ± 0.001	10.57 ± 0.55	0.145 ± 0.025	0.230 ± 0.003	0.523 ± 0.021	4% glycerol YP
GLpm (Δ*mpc1*)	0.165 ± 0.006	10.25 ± 0.49	0.153 ± 0.007	0.278 ± 0.021	0.645 ± 0.062	4% glycerol YP
GLpm	0.058 ± 0.007	9.11 ± 0.72	0.122 ± 0.010	0.143 ± 0.011	0.506 ± 0.009	6% glycerol UAB
GLpm *	0.021 ± 0.003	19.57 ± 3.07	0.163 ± 0.034	0.117 ± 0.025	0.302 ± 0.030	2% glycerol UAB fed−10% glycerol

All the values are means for at least three independent measurements with standard deviation. * In fed-batch, yield and productivity were calculated considering concentrations (g L^−1^) after the feeding pulse at 73 h (t_i_). Subtitle: Y_P/S_: yield of product (lactic acid) per substrate; Y_X/S_: biomass yield per substrate.

## Data Availability

Not applicable.

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
