# Peer review of "Loss of a Functional Mitochondrial Pyruvate Carrier in Komagataella phaffii Does Not Improve Lactic Acid Production from Glycerol in Aerobic Cultivation"

_microorganisms, 2023, doi:10.3390/microorganisms11020483_

Round 1
Reviewer 1 Report
The manuscript is written in good English language. The number of illustrations is sufficient. However, the authors probably did not expect the obtained result and unfortunately could not make a very interesting paper out of this material.
My comments and recommendations.
The title
I recommend the authors to change the title to point out that deletion of the mitochondrial pyruvate transporter does not promote lactate production in K. pfaffii.
Abstract
I suggest that second half of the abstract should be rewritten to state more clearly the conclusions deduced from current study. I feel that the conclusions are largely based on previously published results of the group. If so, it should be indicated.
Line 22 alcohol dehydrogenase two promoter, should be alcohol dehydrogenase II (or 2) promoter.
Introduction
Lines 94-95 …the gene of the bacterial hemoglobin from Vitreoscilla was cloned under the alcohol dehydrogenase 2 promoter [25], known to work as a hypoxia sensor [26]. My comment: the paper under reference no 26 (Melo et al., 2020) has no data concerning this hemoglobin and hypoxia sensing.
I consider that the authors should explain in more detail, how to interpret the data concerning the Vitreoscilla hemoglobin expression. If they have used it earlier to evaluate hypoxia, respective data/results should be discussed.
Figure 1.
I recommend to cross out in Fig.1 the functions that will be eliminated due to gene knockouts. Also, the legend to the figure should indicate the papers where previous genetic modifications of the strain were first published.
Results and discussion
Fig. 3. The timepoint of adding the UK-5099 should be designated on the figure. Also, in panel B, light gray and dark gray colors look quite the same. Question: what about inhibition of the mitochondrial pyruvate transporter by UK-5099 and growth in UAB medium without leucine supplementation?
Fig. 4. Panels A and B are not designated on the figure.
Legend to Fig. 4 (line 279). Growth curves (circles) in 500-mL baffled flasks. Should be… Growth curves of K. pfaffii strains cultivated in 500-mM baffled flasks.
Line 287. Replace greater with higher
Line 282. Carbon source in the medium should be indicated.
Line 326. Fig. 5B. Should be Fig. 5D.
Line 351 and further. Was leucine added to UAB medium when the GLpm strain was cultivated? If yes, it should be noted.
Lines 369-373. The authors state: Both results could be explained if the yeast had a way to circumvent the loss of Mpc1 and shunt pyruvate carbon into the mitochondrion. Conversion to oxaloacetate by pyruvate carboxylase, which can be found in the cytosol in S. cerevisiae [37], followed by mitochondrial uptake via oxaloacetate transporters [38], is a possibility that merits further investigation. My comment: The genome of K. phaffii is available. This assumption can be supported by search/detection of respective genes.
Lines 377-378. Overexpression of hemoglobin in K. phaffii has been reported for other products [28,39]. A bad sentence. Please rephrase.
Author Response
General remarks
You’re right that our results are unexpected and as such, not as interesting as improved lactate yield would’ve been had our work hypothesis been born out by data. However, useful data of non-conventional industrial yeast physiology in specific settings is painfully lacking in bioprocess literature, and our findings both show that even data from closely-related clade member like Saccharomyces cerevisiae cannot be extended to Komagataella phaffii. Given the recent publications on aerobic production of pyruvate-derived compounds in Mpc1 knock-out strains, which were done in flask cultures and using rich media (such as the ones we cite from A. oryzae and S. cerevisiae), our finding that functional hypoxia at high cell densities occurs even with high aeration also contributes to decision-making in scaled bioprocess design for these organisms. All in all, we believe it is in the interest of the yeast research community to add our results to their understanding of K. phaffii.
The title
I recommend the authors to change the title to point out that deletion of the mitochondrial pyruvate transporter does not promote lactate production in K. pfaffii.
We’ve changed the title to “Loss of a functional mitochondrial pyruvate carrier in Komagataella phaffii does not improve lactic acid production from glycerol in aerobic cultivation”
Abstract
I suggest that second half of the abstract should be rewritten to state more clearly the conclusions deduced from current study. I feel that the conclusions are largely based on previously published results of the group. If so, it should be indicated.
Thank you for your suggestion. Please check Abstract.
Line 22 alcohol dehydrogenase two promoter, should be alcohol dehydrogenase II (or 2) promoter.
Ok, done. Please, check L20.
Introduction
Lines 94-95 …the gene of the bacterial hemoglobin from Vitreoscilla was cloned under the alcohol dehydrogenase 2 promoter [25], known to work as a hypoxia sensor [26]. My comment: the paper under reference no 26 (Melo et al., 2020) has no data concerning this hemoglobin and hypoxia sensing.
Thank you for the comment. References were reviewed and corrected. L98
I consider that the authors should explain in more detail, how to interpret the data concerning the Vitreoscilla hemoglobin expression. If they have used it earlier to evaluate hypoxia, respective data/results should be discussed.
Please check L99 to L102.
Figure 1.
I recommend to cross out in Fig.1 the functions that will be eliminated due to gene knockouts. Also, the legend to the figure should indicate the papers where previous genetic modifications of the strain were first published.
Figure 1 and legend were adjusted. Please check L107, L109, L110.
Results and discussion
Fig. 3. The timepoint of adding the UK-5099 should be designated on the figure. Also, in panel B, light gray and dark gray colors look quite the same. Question: what about inhibition of the mitochondrial pyruvate transporter by UK-5099 and growth in UAB medium without leucine supplementation?
Please check new Figure 3. Thank you for your comment. In the proof-of-concept using UK-5099 (Figure 3), the GLp strain - with intact Mpc1 - was cultivated in UAB medium without leucine supplementation. Please check L247.
Fig. 4. Panels A and B are not designated on the figure.
Please check new Figure 4.
Legend to Fig. 4 (line 279). Growth curves (circles) in 500-mL baffled flasks. Should be… Growth curves of K. phaffii strains cultivated in 500-mM baffled flasks.
Ok, done. Please check L291.
Line 287. Replace greater with higher
Ok, done. Please check L300
Line 282. Carbon source in the medium should be indicated.
Ok, done. Please check L293.
Line 326. Fig. 5B. Should be Fig. 5D.
Ok, done. Please check L340.
Line 351 and further. Was leucine added to UAB medium when the GLpm strain was cultivated? If yes, it should be noted.
Thank you for your comment. Leucine was not added in bioreactor cultivations of GLpm. We added a sentence to help clarify this sentence. Please, check L352 L355.
Lines 369-373. The authors state: Both results could be explained if the yeast had a way to circumvent the loss of Mpc1 and shunt pyruvate carbon into the mitochondrion. Conversion to oxaloacetate by pyruvate carboxylase, which can be found in the cytosol in S. cerevisiae [37], followed by mitochondrial uptake via oxaloacetate transporters [38], is a possibility that merits further investigation. My comment: The genome of K. phaffii is available. This assumption can be supported by search/detection of respective genes.
We had blasted the relevant S. cerevisiae sequences against the K. phaffii genome even before the project started, but felt reporting just the fact we had high-similarity hits was of little interest for the more speculative section of our discussion. However, your comment has led us to reassess that judgement and we have added the relevant GenBank data, thank you for that. Please check lines 469-74.
Lines 377-378. Overexpression of hemoglobin in K. phaffii has been reported for other products [28,39]. A bad sentence. Please rephrase.
Thank you for your suggestion. Please check L394, 395.
Reviewer 2 Report
Please check the attached file.

Author Response
Overall, ”.” and ”,” are used inappropriately.
For example:
L88: 64,6% → 64.6%
Thank you. Punctuations were reviewed throughout the manuscript.
Numbers on the vertical axis of Figure 3 B
Ok, done. Please check L251
L250-251: 9,0 ± 7,0 µM → 9.0 ± 7.0 µM
Ok, done. L261
L286: 1,32 times → 1.32 times
Ok, done. L299
Numbers on the vertical axis of Figure 5 C and Figure 5 D
Ok, done. Please check L326
Numbers on the vertical axis of Figure 6 C and Figure 6 D and so on.
Ok, done. Please check L372
Please read carefully throughout.
L165: pH 5 → pH 5.0 L173: pH was maintained at 5 → pH was maintained at 5.0
Ok, done. Please check L164, L172, L175, and L181.
L246: 1,5 g.L-1 → 1.5 g L -1 I don't think the "." in " g.L-1 " is necessary. Throughout this paper, it is written as " g.L-1 ".
Punctuations were reviewed throughout the manuscript. Please check L216, L217, L219, L258, L264, L265, and Table 3.
There is no notation of A and B in the graph of Figure 4.
Please check new Figure 4, L290.
Round 2
Reviewer 1 Report
The manuscript looks much better now. I still recommend to reformulate the conclusion (lines 419-422). ... Given that the loss of Mpc1 would reduce the metabolic performance of the yeast under aerobiosis, this requirement for oxygen suggests it might be challenging to use aerobiosis to improve yields in a bioreactor setting. This sentence does not say much and is confusing.
The abstract (lines 32-33) says...overall, the results suggest that blocking cell respiration using hypoxia is more suitable than deleting Mpc for producing lactic acid in K. phaffii. I recommend to rephrase it showing that this conclusion is based not only on current work, but also on earlier relevant studies of the group.
Author Response
Abstract
The abstract (lines 32-33) says...overall, the results suggest that blocking cell respiration using hypoxia is more suitable than deleting Mpc for producing lactic acid in K. phaffii. I recommend to rephrase it showing that this conclusion is based not only on current work, but also on earlier relevant studies of the group.
Thank you for your suggestion. We have added a short phrase to that extent. Please check line 27.
Results and discussion
I still recommend to reformulate the conclusion (lines 419-422). ... Given that the loss of Mpc1 would reduce the metabolic performance of the yeast under aerobiosis, this requirement for oxygen suggests it might be challenging to use aerobiosis to improve yields in a bioreactor setting. This sentence does not say much and is confusing.
We have attempted to flesh this sentence out by explaining there are no gains in lactic acid yield in the knockout strain, and that the deletion causes the yeast to grow a little slower. We also explain how our results show that aerobic production of lactic acid fails to outdo the hypoxic settings as we intended, and that the latter thus remains as the paradigm for this bioprocess unless completely new strategies are developed. Please check lines 503-509.